# DUMB: A Benchmark for Smart Evaluation of Dutch Models

**Wietse de Vries**
University of Groningen
The Netherlands
wietse.de.vries@rug.nl

**Martijn Wieling**
University of Groningen
The Netherlands
m.b.wieling@rug.nl

**Malvina Nissim**
University of Groningen
The Netherlands
m.nissim@rug.nl

## Abstract

We introduce the Dutch Model Benchmark: DUMB. The benchmark includes a diverse set of datasets for low-, medium- and high-resource tasks. The total set of nine tasks includes four tasks that were previously not available in Dutch. Instead of relying on a mean score across tasks, we propose Relative Error Reduction (RER), which compares the DUMB performance of language models to a strong baseline which can be referred to in the future even when assessing different sets of language models. Through a comparison of 14 pre-trained language models (mono- and multi-lingual, of varying sizes), we assess the internal consistency of the benchmark tasks, as well as the factors that likely enable high performance. Our results indicate that current Dutch monolingual models under-perform and suggest training larger Dutch models with other architectures and pre-training objectives. At present, the highest performance is achieved by DeBERTaV3$_{large}$, XLM-R$_{large}$ and mDeBERTaV3$_{base}$. In addition to highlighting best strategies for training larger Dutch models, DUMB will foster further research on Dutch. A public leaderboard is available at dumbench.nl.

## 1 Introduction

To evaluate and compare new and existing language models, a reliable method of comparing model quality is essential. For this reason, several benchmark suites have been proposed, such as English GLUE (Wang et al., 2018) and SuperGLUE (Wang et al., 2019). However, at present, there is no standard benchmark for Dutch.

The currently available Dutch language models are BERTje (de Vries et al., 2019), which is a Dutch version of BERT$_{base}$ (Devlin et al., 2019a), and three versions of RobBERT (Delobelle et al., 2020, 2022b) which are Dutch versions of RoBERTa$_{base}$ (Liu et al., 2019). Direct comparisons between these models have focused on

several tasks: sentiment analysis, natural language inference, coarse-grained part-of-speech tagging and three-class named entity recognition, where RobBERT often outperforms BERTje (Delobelle et al., 2022b). Results are not completely consistent, however. Some studies found that Rob-BERT performs better than BERTje at specific tasks (Ruitenbeek et al., 2022; Delobelle et al., 2022b; De Bruyne et al., 2021), whereas other studies find the opposite (Wijnholds and Moortgat, 2021; De Langhe et al., 2022). Other works assume higher performance for specific models and either exclusively experiment with BERTje (Alam et al., 2021; Ghaddar et al., 2021; Brandsen et al., 2022), or exclusively experiment with RobBERT (Spruit et al., 2022; Delobelle et al., 2022a).

By developing a new benchmark, we aim to reduce the present unclarity of current evaluations and obtain insights into potential performance improvements for future development of new Dutch models. We also hope that this work will foster further research on Dutch, including the development of decoder-based models. Indeed, we see the establishment of such a benchmark, together with the evaluation of existing encoder-based models, as a necessary step towards making it possible to also devise a solid evaluation framework for generative models, which is complicated by the high degree of variability related to prompts and outputs.

**Contributions**

- We create a balanced Dutch benchmark (DUMB) of nine tasks, including four which were previously unavailable.

- We propose using a Relative Error Reduction (RER) metric to compare relative model performance across tasks.

- We assess the current state of Dutch language modeling, also through comparison against multilingual and English models.

- We identify and quantify limitations of current Dutch models and propose directions for potential developments.

**Other Benchmarks**

Various monolingual and multilingual benchmarks exist. For English, the two standard benchmarks are GLUE (Wang et al., 2018) and SuperGLUE (Wang et al., 2019). Four of the nine tasks in GLUE are comparable Natural Language Inference (NLI) tasks, and three of the remaining tasks are semantic similarity tasks. SuperGLUE has a focus on Question Answering (QA), with 4 of the 8 tasks being QA tasks.

Multiple efforts exist to make GLUE-like benchmarks for other languages, such as (Chinese) CLUE (Xu et al., 2020), BasqueGLUE (Urbizu et al., 2022), RussianSuperGLUE (Shavrina et al., 2020), (Korean) KLUE (Park et al., 2021), (French) FLUE (Le et al., 2020), (Japanese) JGLUE (Kurihara et al., 2022), (Indonesian) IndoNLU (Wilie et al., 2020) and (Arabic) ORCA (Elmadany et al., 2022). These benchmarks contain varying numbers of tasks relating to, for instance, NLI, QA, Semantic Textual Similarity (STS), Word Sense Disambiguation (WSD). All monolingual benchmarks use the mean task score as an aggregate measure. We discuss in Section 3.3 why this might not be optimal, and suggest an alternative approach for comparative evaluation. Reported baseline experiments sometimes include only monolingual models of the target language (Park et al., 2021; Urbizu et al., 2022; Elmadany et al., 2022). For other benchmarks, base-sized multilingual models (Shavrina et al., 2020) or base- and large-sized multilingual models are included as well (Kurihara et al., 2022; Wilie et al., 2020). For the latter two studies, the *large* multilingual models outperform the (*base*-sized) monolingual models.

Multilingual benchmarks have also been proposed. XGLUE (Liang et al., 2020), XTREME (Hu et al., 2020) and XTREME-R (Ruder et al., 2021) cover 19, 40 and 50 languages, respectively. Dutch is included in these benchmarks, but only for coarse-grained Universal Dependencies Part-Of-Speech (POS) tagging (Zeman et al., 2022) and automatically generated WikiANN Named Entity Recognition (Pan et al., 2017). These multilingual benchmarks only contain English training data, and are therefore tailored to evaluate cross-lingual transfer rather than monolingual performance.

| Task | Dataset | M. | C. | \|Train\| | \|Dev\| | \|Test\| |
|------|---------|-----|----|----------|---------|----------|
| POS | Lassy | acc. | 218 | 59,167 | 1,814 | 4,184 |
| NER | SoNaR-1 | F1 | 7 | 54,472 | 1,392 | 4,080 |
| WSD | WiC-NL | acc. | 2 | 7,184 | 1,330 | 1,330 |
| PR | DPR | acc. | 2 | 786 | 142 | 1,216 |
| CR | COPA-NL | acc. | 2 | 400 | 100 | 500 |
| NLI | SICK-NL | acc. | 3 | 4,439 | 495 | 4,906 |
| SA | DBRD | acc. | 2 | 19,528 | 500 | 2,224 |
| ALD | DALC | F1 | 3 | 6,817 | 1,205 | 3,270 |
| QA | SQuAD-NL | F1 | - | 130,319 | 10,174 | 1,699 |

Table 1: The nine DUMB tasks and their evaluation **M**etrics, number of **C**lasses, and split sizes. Underlined datasets (WSD, PR, CR, QA) and splits (POS, NER, WSD, SA, QA) are newly introduced in this paper.

## 2  DUMB Tasks

A general language model should be able to perform well at different types of tasks. Therefore, we balance our benchmark to contain word-level, word-pair-level, sentence-pair level and document-level tasks. Moreover, we include tasks having different orders of magnitude of training items. We call these low-resource ($10^2$), mid-resource ($10^3$), and high-resource ($>10^4$) tasks.

All included datasets are freely available and most source datasets use permissive licenses that allow (modified) redistribution. We do however not currently share the pre-processed benchmark directly because of licensing restrictions of some datasets. Simple instructions to download these datasets and a single pre-processing script are available on our Github so that the entire benchmark can be obtained. The included tasks are listed in Table 1 and described below. Appendix A contains example items for each task.

### 2.1  Word Tasks

Word tasks involve classifying or tagging individual words. The surrounding sentence can be used to determine lexicosemantic classes for individual (compound) words.

**Part-Of-Speech Tagging (POS)** We use the fine-grained POS annotations from the Lassy Small v6.0 corpus (van Noord et al., 2013). The corpus permits non-commercial use (custom license). This data was annotated using the Corpus Gesproken Nederlands (CGN) guidelines that define 316 distinct morphosyntactic tags (van Eynde, 2004), of which 218 tags are used in the Lassy corpus.

The source corpus contains documents from magazines, newsletters, web pages, Wikipedia, press releases, novels, brochures, manuals, legal texts and reports. Only the Wikipedia section of this corpus is included in Universal Dependencies (Zeman et al., 2022). For our benchmark, we introduce document-level random cross validation splits. We reserve 2% of documents as development data, 5% as test data and the remaining 93% as training data. We selected the random seed such that all 218 classes appear in the training data.

**Named Entity Recognition (NER)**   For NER, we use the SoNaR-1 v1.2.2 corpus (Oostdijk et al., 2013). The corpus permits non-commercial use (custom license). In addition to the person, organization, location and miscellaneous entity types of CoNLL-2002 (Tjong Kim Sang, 2002), SoNaR-1 contains separate classes for products and events, resulting in 6 entity classes and a negative class.

The SoNaR-1 source data largely overlaps with the POS source data described in the previous section and contains the same domains. Cross validation splits were made in identical fashion. To facilitate transfer or multi-task learning, we ensure that documents that have annotations for both tasks are in the same splits for both tasks.

## 2.2   Word Pair Tasks

Word pair tasks involve comparison of words or small clusters of words. These tasks are specifically designed to test disambiguation.

**Word Sense Disambiguation (WSD)**   As our Word Sense Disambiguation (WSD) task, we replicate the English Words in Context (WiC; Pilehvar and Camacho-Collados, 2019) task. WiC items contain two sentences from different contexts that contain the same target word. The task is a binary classification task to determine whether the words share the same sense in both sentences.

For English WiC, the word senses of WordNet (Miller, 1994) are used to determine sense equivalence (Pilehvar and Camacho-Collados, 2019). The used sentences were extracted from WordNet, VerbNet (Kipper Schuler et al., 2009) and Wiktionary. This English task is included in the SuperGLUE benchmark (Wang et al., 2019). A multilingual version of WiC, XL-WiC (Raganato et al., 2020), has been constructed based on WordNet and Wiktionary in other languages. This dataset contains Dutch test data that has been extracted

from Open Dutch WordNet (Postma et al., 2016), but it is lacking Dutch training data.

To replicate WiC and XL-WiC for Dutch (WiC-NL), we use the sense-tagged data from DutchSemCor (Vossen et al., 2012). The source dataset allows modification and redistribution (CC-BY 3.0). DutchSemCor provides Cornetto (Vossen et al., 2008) word sense annotations for documents from the SoNaR-500 corpus (Oostdijk et al., 2013). We extract and group sentences by POS tag (adjectives, nouns and verbs), lemmas and word senses. For each lemma, we randomly select an equal number of positive and negative pairs of sentences, where the word sense is either the same or different. Note that we do this by lemma, so the target word may have different morphology in the two sentences. For cross validation, we split our dataset by lemma, so none of the words in the test data have been seen during training. The same lemma appears at most six times in training and four times in development and test data. The development and test data each contain 15% of the total set of lemmas. The final dataset contains 1.3K development and test items and 7.2K train items. This size is similar to English WiC (5.4K train, 0.6K development and 1.4K test).

As positive and negative classes are balanced, majority and random performance is 50%. State-of-the-art English WiC accuracy is 77.9% (T5 + UDG; Wang et al., 2021) and the highest cross-lingual Dutch XL-WiC accuracy is 72.8 (XLM-$R_{large}$; Raganato et al., 2020). We expect WiC-NL performance at a similar order of magnitude.

**Pronoun Resolution (PR)**   Similar to the Winograd Schema Challenge (Levesque et al., 2012) that is included in SuperGLUE, we include a pronoun resolution task in our benchmark. We use coreference annotations from SemEval2010 Task 1 (Recasens et al., 2010) as source data to construct a Dutch Pronoun Resolution (DPR) dataset. The dataset permits non-commercial use (custom license). We do not directly redistribute the data, but do automate pre-processing. We cast this task as a balanced binary classification with one or two sentences as input. In the input, a pronoun and a non-pronoun entity are marked and the task is to determine whether the pronoun refers to the entity. Entities in negative samples always refer to a different entity or pronoun.

Cross validation splits are taken from the source SemEval data. The final dataset contains only 768

training items, which makes this task relatively low-resource. Positive and negative cases are balanced, so majority and random accuracy is 50%.

## 2.3 Sentence Pair Tasks

Sentence pair tasks test whether models recognize semantic relationships between sentences. Specifically, we test temporal causal relationships and entailment relationships.

**Causal Reasoning (CR)** We have translated the Choice of Plausible Alternatives (COPA; Gordon et al., 2012) dataset to Dutch (COPA-NL). COPA is distributed under the 2-Clause BSD License and permits modified redistribution. In this causal reasoning task, one of two causally related sentences has to be selected based on a premise. We used Google Translate (Nov. 2022) to translate all items, and manually corrected translation errors for the entire dataset in two passes: (i) we checked that individual sentences were correct translations, and (ii) we assessed the coherence of sentence pairs and corresponding labels.

The train, dev and test splits are 400, 100 and 500 items, respectively. Due to the limited training data size, English models typically use auxiliary training data such as the Social IQa dataset (Sap et al., 2019). With auxiliary data, English COPA accuracy can reach 98.4% (He et al., 2021). In our experiments, we will not use any auxiliary data, so performance is expected to be lower. XCOPA, a translation of COPA in 11 other languages (Ponti et al., 2020), can reach 55.6% average accuracy over all languages and 69.1% when using auxiliary training data (XLM-R; Conneau et al., 2020).

**Natural Language Inference (NLI)** We use SICK-NL (Wijnholds and Moortgat, 2021) for NLI, a Dutch translation of SICK (Marelli et al., 2014), distributed under the permissive MIT license. This is a three-class NLI sentence pair classification task (entailment, contradiction, neutral). SICK also includes human judgements of semantic textual similarity (STS), but this task is not part of DUMB in order to preserve task type balance. For SICK-NL, accuracies up to 84.9% have been reported (RobBERT$_{2022}$; Delobelle et al., 2022b).

## 2.4 Document Tasks

Document tasks involve classification of a multi-sentence text and extractive question answering.

**Sentiment Analysis (SA)** We use the Dutch Book Reviews Dataset v3.0 (DBRD; Van der Burgh and Verberne, 2019), which is distributed with the permissive MIT license. This task involves classifying a book review as positive or negative. We remove 500 reviews from the training set to use for development, since the original data is missing a dev set. The highest previously reported accuracy on this task (with original splits) is 95.1% (RobBERT$_{2022}$; Delobelle et al., 2022b).

**Abusive Language Detection (ALD)** We include an abusive language detection task based on DALC v2.0 (Ruitenbeek et al., 2022), which is distributed with the permissive GPLv3 license. This dataset contains annotations for anonymized abusive and offensive Twitter messages. The specific task we include is a three-way classification of abusive, offensive or neutral tweets. The highest previously achieved macro F1 score of this task is 63.7% (RobBERT$_{V2}$; Ruitenbeek et al., 2022).

**Question Answering (QA)** We translated SQuAD2.0 (Rajpurkar et al., 2016, 2018) to Dutch using Google Translate (Feb. 2023) with post-editing. SQuAD-NL consists of Wikipedia paragraphs and questions for which the answer can be found in the context paragraph. Unanswerable questions are also included. As the original SQuAD test-data is not public, we use the same 240 paragraphs that were selected in XQuAD (Artetxe et al., 2020) as test data, and the remaining 1,827 paragraphs are used as our development data. The test data was manually corrected by eight BSc students as part of their thesis work. For SQuAD-NL we use the same license of the original dataset (CC-BY-SA 4.0). We also distribute SQuAD-NL version 1.1, which does not contain unanswerable questions.

## 3 Evaluation

We conduct an analysis of several pre-trained language models (PLMs) with DUMB. The goal of this analysis is to identify strengths and limitations of current Dutch language models, as well as to determine the overall effectiveness of the proposed benchmark. We evaluate the influence of model variants, model sizes and pre-training languages.

## 3.1 Pre-trained Language Models

The model types we include in our evaluation include three Transformer-encoder (Vaswani et al.,

2017) variants, namely BERT (Devlin et al., 2019a), RoBERTa (Liu et al., 2019) and DeBER-TaV3 (He et al., 2023). For each model, we fine-tune the *base* model size and the *large* model size, if available. Models with the *base* and *large* model sizes have 85M and 302M trainable parameters in their Transformer-encoder layers, respectively. Due to varying vocabulary sizes, the total word embedding layers vary in size.

All currently available Dutch monolingual language models are included in our experiments, as well as their multilingual equivalents. Dutch is included in the set of pre-training languages of each multilingual model. We also include the monolingual English model variants since they have been shown to transfer well to non-English languages (Artetxe et al., 2020; Blevins and Zettlemoyer, 2022). Language models can in general transfer well to other languages through either cross-lingual fine-tuning a multilingual model (de Vries et al., 2022), or through various continual pre-training approaches (Gogoulou et al., 2022; de Vries and Nissim, 2021; Li et al., 2021). However, monolingual English models can also perform well on non-English languages by merely fine-tuning in the target language (Blevins and Zettlemoyer, 2022). Monolingual transfer effectiveness is facilitated by language contamination during pre-training, which is higher for RoBERTa than for BERT (Blevins and Zettlemoyer, 2022).

As BERT-type models, we take English BERT (Devlin et al., 2019a), cased multilingual BERT (mBERT; Devlin et al., 2019b) and Dutch BERTje (de Vries et al., 2019). These are all pre-trained on curated data with Masked Language Modeling (MLM) and the Next Sentence Prediction (NSP) or Sentence Order Prediction (SOP) objectives. These models are the first published English, multilingual and Dutch language models, respectively. A *large* variant is only available for English.

As RoBERTa-type models, we include English RoBERTa (Liu et al., 2019), multilingual XLM-RoBERTa (XLM-R; Conneau et al., 2020) and three versions of RobBERT (Delobelle et al., 2020, 2022b). These models are pre-trained with only the MLM objective, and their pre-training datasets have larger sizes and lower curation due to the inclusion of web scraped data. The Dutch RobBERT$_{V1}$ and RobBERT$_{V2}$ models are trained with the OSCAR 2019 corpus (Ortiz Suárez et al., 2020); the first version used the original English byte-pair-encoding vocabulary, while the second used a new Dutch vocabulary. The RobBERT$_{2022}$ update is trained with the same procedure as V2, but with the larger OSCAR 22.01 corpus (Abadji et al., 2022). Large model variants are only available for English and multilingual RoBERTa.

As DeBERTaV3-type models, we include English DeBERTaV3 (He et al., 2023) and multilingual mDeBERTaV3 (He et al., 2023). DeBERTaV3 primarily differs from BERT and RoBERTa by disentangling content and position embeddings (He et al., 2021). Moreover, DeBERTaV3 and mDeBERTaV3 are pre-trained with an ELECTRA-style (Clark et al., 2020) generator-discriminator training with gradient-disentangled word embeddings (He et al., 2023). DeBERTaV3 and mDeBERTaV3 outperform RoBERTa and XLM-RoBERTa on GLUE (Wang et al., 2018) and XNLI (Conneau et al., 2018), respectively, despite being pre-trained with the same data. A *large* model variant is only available for English.

## 3.2 Fine-tuning Procedure

Our benchmark contains several types of tasks requiring different implementations. Reference implementations in the Hugging Face Transformers library for each task can be found in our repository.[1] Specifically, we provide an implementation for token classification (POS, NER), span classification (WSD, PR), multiple choice (CR), sequence classification (NLI, SA, ALD), and extractive question answering (QA).

We fine-tune each of the pre-trained models on the tasks with individual hyper-parameter grid-searches for each model and task. Optimal hyper-parameters are chosen based on validation data, and differ between models and tasks. We optimize numbers of epochs, warm-up steps, learning rate and dropout. After the hyper-parameter search, we rerun fine-tuning with 5 different random seeds. Reported scores are average test data performance of those 5 runs. Grid search ranges, optimal hyper-parameters, and training durations are in Appendix B. In our baseline experiments, we fine-tune the pre-trained models on the benchmark task training data without exploring transfer from similar tasks or special sampling techniques.

---

[1]`github.com/wietsedv/dumb/tree/main/trainers`

| | Avg | Word | | | | Word Pair | | | | Sentence Pair | | | | Document | | | | | |
|---|---|---|---|---|---|---|---|---|---|---|---|---|---|---|---|---|---|---|---|
| | | POS | | NER | | WSD | | PR | | CR | | NLI | | SA | | ALD | | QA | |
| Model | RER | RER | Acc. | RER | F1 | RER | Acc. | RER | Acc. | RER | Acc. | RER | Acc. | RER | Acc. | RER | F1 | RER | F1 |
| 🇳🇱 BERTje | 0 | 0 | 97.8 | 0 | 86.1 | 0 | 65.9 | **0** | 65.8 | 0 | 62.0 | 0 | 85.2 | 0 | 93.3 | 0 | 58.8 | 0 | 70.3 |
| 🇳🇱 RobBERT$_{V1}$ | -16.3 | 12.5 | 98.1 | -19.4 | 83.5 | -15.3 | 60.6 | -24.0 | 57.6 | -14.7 | 56.4 | -12.7 | 83.3 | -58.2 | 89.4 | 4.8 | 60.8 | -19.4 | 64.6 |
| 🇳🇱 RobBERT$_{V2}$ | 1.6 | 16.2 | 98.2 | 4.1 | 86.7 | -5.3 | 64.1 | **0.1** | **65.8** | -10.2 | 58.1 | -3.8 | 84.6 | -0.5 | 93.2 | 12.0 | 63.7 | 2.2 | 71.0 |
| 🇳🇱 RobBERT$_{2022}$ | 3.6 | 17.3 | 98.2 | 7.6 | 87.2 | -6.4 | 63.7 | **-1.8** | **65.2** | -10.1 | 58.2 | 3.1 | 85.6 | 4.0 | 93.5 | 18.9 | 66.6 | -0.2 | 70.3 |
| 🌐 mBERT$_{cased}$ | -5.8 | 6.2 | 97.9 | 9.2 | 87.4 | 7.7 | 68.5 | -11.0 | 62.0 | -18.4 | 55.0 | -6.2 | 84.3 | -41.7 | 90.5 | -4.5 | 56.9 | 6.9 | 72.4 |
| 🌐 XLM-R$_{base}$ | -0.3 | 13.9 | 98.1 | 10.8 | 87.6 | 1.9 | 66.5 | -16.2 | 60.2 | -26.8 | 51.8 | 2.0 | 85.5 | -3.6 | 93.0 | 3.4 | 60.2 | 12.3 | 74.0 |
| 🌐 mDeBERTaV3$_{base}$ | 12.8 | 18.2 | 98.2 | 17.2 | 88.5 | 10.8 | 69.6 | -20.8 | 58.7 | 19.7 | 69.5 | **25.2** | **88.9** | 3.3 | 93.5 | 12.4 | 63.9 | 29.2 | 79.0 |
| 🌐 XLM-R$_{large}$ | 14.4 | **26.5** | **98.4** | **29.7** | **90.3** | **21.3** | **73.1** | -15.8 | 60.4 | -25.8 | 52.2 | 24.4 | 88.8 | **13.2** | **94.2** | **19.0** | **66.6** | 37.2 | 81.4 |
| 🇺🇸 BERT$_{base}$ | -42.8 | -19.8 | 97.4 | -30.8 | 81.9 | -22.4 | 58.2 | -18.7 | 59.4 | -28.0 | 51.4 | -19.2 | 82.3 | -203.9 | 79.6 | -16.1 | 52.2 | -26.2 | 62.5 |
| 🇺🇸 RoBERTa$_{base}$ | -25.6 | -6.5 | 97.7 | -27.3 | 82.3 | -14.0 | 61.1 | -20.4 | 58.8 | -24.1 | 52.8 | -19.7 | 82.3 | -99.9 | 86.6 | -16.0 | 52.2 | -2.1 | 69.7 |
| 🇺🇸 DeBERTaV3$_{base}$ | -1.6 | 6.5 | 97.9 | 1.7 | 86.4 | -4.2 | 64.4 | -25.3 | 57.1 | -20.5 | 54.2 | 8.6 | 86.5 | -14.6 | 92.3 | 3.5 | 60.2 | 29.7 | 79.1 |
| 🇺🇸 BERT$_{large}$ | -35.1 | -12.0 | 97.5 | -25.9 | 82.5 | -25.4 | 57.2 | -29.3 | 55.8 | -31.2 | 50.2 | -15.4 | 82.9 | -158.7 | 82.6 | -7.8 | 55.6 | -10.4 | 67.2 |
| 🇺🇸 RoBERTa$_{large}$ | -14.1 | 6.4 | 97.9 | -12.3 | 84.4 | -19.8 | 59.1 | -23.3 | 57.8 | -26.1 | 52.1 | -8.5 | 83.9 | -63.8 | 89.0 | 1.2 | 59.3 | 19.7 | 76.2 |
| 🇺🇸 DeBERTaV3$_{large}$ | 15.7 | 17.9 | 98.2 | 10.9 | 87.6 | 12.7 | 70.2 | -14.4 | 60.9 | **35.4** | **75.4** | 24.1 | 88.7 | -6.4 | 92.8 | 12.5 | 64.0 | **48.4** | **84.7** |

Table 2: Task scores and Relative Error Reduction (RER) scores per model. Models are grouped by pre-train language and model size. Bold values indicate highest (or not significantly different, $p \geq 0.05$) scores per task. Gray values are significantly ($p < 0.05$) below baseline. Significance testing is described in Section 3.3. Updated results with newer models can be found on `dumbench.nl`.

## 3.3 Evaluation Method

We provide a single aggregate score based on all tasks. In existing benchmarks, the arithmetic mean score of the different tasks is used, despite varying metrics and task difficulties within the benchmark (Wang et al., 2018, 2019; Hu et al., 2020). This assumes that absolute score differences are equally meaningful across tasks, which could be the case if the set of tasks is relatively homogeneous. However, our set has a high variation in expected scores with around 95% accuracy on POS and only 70% on CR. We assume that a single point improvement for POS (effectively reducing the error by 20%) would then be more meaningful than a single point improvement on CR (effectively reducing the error by about 3%).

As a solution, we propose to not use the mean score per model, but the average Relative Error Reduction (RER) compared to a strong baseline. For instance if the baseline task performance is 80% and a target model achieves 85%, RER score is $5\%/20\% = 25\%$. For our baseline model, we choose the Dutch BERTje model. This is the first Dutch pre-trained language model and is only available in *base* size. We argue that any newer and larger models should outperform this baseline to be considered useful for practical applications.

To evaluate whether language models do not significantly differ from the best performing model per task, or perform significantly lower than the baseline model per task, we fit two binomial mixed effects regression models per task. Correctness of the each item is used as dependent variable in all cases, and by-item random intercepts are included to account for the item-based variability. The predictor of interest is the model (i.e. a 14-level nominal variable). The two regression models per task use the baseline model or the best performing model as reference levels. Consequently, for each item 70 predictions are included (14 models times five runs per model) in all 18 regression models (two regression models for each task). We use a significance threshold $\alpha$ of 0.05 in evaluating the $p$-values distinguishing the performance of each model from that of the chosen reference level.

| | POS | NER | WSD | PR | CR | NLI | SA | ALD | QA |
|---|---|---|---|---|---|---|---|---|---|
| **POS** | - | 0.85 | 0.75 | 0.31 | 0.43 | 0.77 | **0.89** | **0.93** | 0.66 |
| **NER** | 0.85 | - | **0.92** | 0.41 | 0.42 | **0.88** | 0.87 | 0.81 | 0.75 |
| **WSD** | 0.75 | **0.92** | - | 0.35 | 0.52 | 0.86 | 0.77 | 0.64 | 0.75 |
| **PR** | 0.31 | 0.41 | 0.35 | - | 0.29 | 0.15 | 0.50 | 0.38 | -0.03 |
| **CR** | 0.43 | 0.42 | 0.52 | 0.29 | - | 0.64 | 0.48 | 0.47 | 0.51 |
| **NLI** | 0.77 | 0.88 | 0.86 | 0.15 | **0.64** | - | 0.74 | 0.79 | **0.87** |
| **SA** | 0.89 | 0.87 | 0.77 | **0.50** | 0.48 | 0.74 | - | 0.82 | 0.66 |
| **ALD** | **0.93** | 0.81 | 0.64 | 0.38 | 0.47 | 0.79 | 0.82 | - | 0.59 |
| **QA** | 0.66 | 0.75 | 0.75 | -0.03 | 0.51 | 0.87 | 0.66 | 0.59 | - |
| | 0.70 | 0.74 | 0.70 | 0.30 | 0.47 | 0.71 | 0.72 | 0.68 | 0.59 |

Table 3: Correlations between tasks (RER) for the models in Table 2. Highest correlations per column are shown in bold and correlations that are not significant (based on 14 values) are shown in gray ($p < 0.05$).

## 3.4 Results

Results for each model and task, grouped by pre-training language and model size, are in Table 2. For model comparison we consider RER per task and average RER across tasks as the most important metrics, but we also report the conven-

tional metrics for each task. Highest overall performance is achieved by DeBERTaV3$_{large}$, an English model. In the following sections we compare performance across tasks and per model, and discuss patterns regarding model types, sizes and pre-training languages.

## 4 Analysis of Tasks

Table 2 shows great variation across tasks, with baseline scores between 58.8% (ALD) and 97.8% (POS). Since absolute scores are not comparable across tasks and metrics, we run a Pearson correlation between RER scores of each pair of tasks (Table 3) to analyse how tasks relate to each other.

We use cross-task correlations as a sanity check for our benchmark. Positive correlations are expected between all tasks and strong correlations between similar tasks, because all tasks should achieve higher performance if the Dutch language is better represented by a model. The word-level tasks (POS and NER) and the document-level tasks (SA and ALD) have strong pairwise correlations of 0.84 and 0.82, respectively. Correlations above 0.9 are observed between POS and ALD (0.93), NER and WSD (0.92). For NER and WSD this makes sense since both tasks involve analysis of syntactic and semantic information of content words. For POS and ALD the results are less intuitive. Given the low absolute scores of ALD (maximum F1 is 66.6), we hypothesize that the language models converge to a non-optimal solution that relies more on individual words than on broader context. Moreover, the correlation between tasks seems more closely related to the training set sizes than task similarity.

### 4.1 High- and Low-resource Tasks

The two low-resource tasks (PR, CR) show the weakest average correlations of 0.30 and 0.47, which suggests that training sizes might contribute to model performance differences. Three high-resource tasks (POS, NER, SA) strongly correlate with each other, with correlations ranging between 0.85 and 0.89, but these tasks correlate less strongly with QA, the highest-resource task. Mid- and high-resource tasks show average correlations of 0.59 (QA) to 0.74 (NER) and five of these seven tasks have correlations of are at least 0.70.

The two low-resource tasks PR and CR do not correlate with other tasks except for a single significant correlation between CR and NLI. This correlation makes sense, since both tasks involve evaluating causality between two sentences. CR is the lowest resource task, and only multilingual mDeBERTaV3 and English DeBERTaV3$_{large}$ manage to outperform the BERTje baseline. These two models also perform especially well on the closely related NLI task, for which they achieve best performance across models.

*Large* variants of English BERT and RoBERTa reach lower PR and CR performance than *base* variants (Table 2). Moreover, these two tasks have the lower RER scores across tasks for multilingual models (except mBERT SA). This suggests that non-Dutch PLMs, and especially larger variants have a disadvantage at low-resource tasks. However, English and multilingual CR performance is still high for DeBERTaV3-based models. Relatively stable Dutch monolingual and high DeBERTaV3 performances indicate that it is not impossible to achieve good performance on small datasets. Since any model could in theory achieve high performance given enough data, we consider it important that models generalize well with little data.

High DeBERTaV3 performance can be related to its pre-training procedure with the ELECTRA-style objective and Gradient Disentangled Embedding Sharing (GDES). To verify this, we tested the English DeBERTa$_{large}$ model, which is not included in Table 2 due to lack of a multilingual variant. This model was pre-trained using Masked Language Modeling, but is architecturally identical to DeBERTaV3$_{large}$. English DeBERTa$_{large}$ achieves -33.7 average RER, and -31.6 and 0.0 RER on CR and NLI, as opposed to 35.4 and 24.1 with DeBERTaV3$_{large}$, suggesting that high cross-lingual DeBERTaV3 performance is primarily due to ELECTRA-style pre-training and GDES.

| | Dutch | | Multi | | English | |
|---|---|---|---|---|---|---|
| | *base* | *large* | *base* | *large* | *base* | *large* |
| BERT | 0 | 4.3 $^{9.6}$ | -5.8 | 2.8 $^{8.1}$ | -42.8 | -35.1 |
| RoBERTa | 3.6 | 13.4 $^{7.8}$ | -0.3 | 14.4 | -25.6 | -14.1 |
| DeBERTaV3 | 24.1 $^{8.1}$ | 38.0 $^{10.8}$ | 12.8 | 36.4 $^{8.6}$ | -1.6 | 15.7 |

Table 4: Mean RER scores for different model types, sizes, and pre-training languages. The Dutch RoBERTa variant is RobBERT$_{2022}$. Black scores correspond to the average RER scores in Table 2. Estimated scores from a linear regression model are shown in gray, with standard errors as superscripts. Note that standard errors are high, since the estimates are based on only 12 observations.

# 5 Analysis of Models

Table 2 shows the results of models of varying types, sizes, and pre-training languages. Surprisingly, the monolingual English DeBERTaV3$_\text{large}$ model achieves highest overall performance, mostly thanks to outstanding performance on the CR and QA tasks. The other two high-performing models are the multilingual mDeBERTaV3 and XLM-R$_\text{large}$ models. In this section we discuss how model types, sizes and pre-training languages compare and why Dutch models are outperformed by non-Dutch models.

To estimate the effects of pre-train languages, model types and model sizes, we fit a linear regression model with those variables as predictors (see Appendix C). A model with all three predictors fits the data significantly better than using single predictors (model comparison: $p < 0.05$) and explains 83.6% of variance (adjusted $R^2$: 73.3%). All three predictors significantly contribute to this model ($p < 0.05$); we discuss them in the following subsections. Interactions between predictors did not significantly improve the model fit.

The average RER scores are shown in Table 4, where missing values are estimated by the regression model. According to this analysis, a Dutch DeBERTaV3$_\text{large}$ model could potentially achieve an error reduction which is more than two times higher than the current best model (38.0 vs. 15.7).

## 5.1 Pre-training Language

Dutch monolingual models seem to outperform equivalent multilingual models and multilingual models outperform equivalent English models. According to the regression model, language accounts for a 1.6 point (non-significant, $p = 0.83$) decrease for multilingual pre-training compared to Dutch monolingual pre-training and a 28.9 point decrease for English pre-training compared to Dutch monolingual pre-training. On the basis of these results, we cannot conclusively claim that current monolingual Dutch models are preferable over multilingual models. Table 2 does however clarify that Dutch models outperform equivalent multilingual models for all tasks except POS, NER and QA. Multilingual and English models perform particularly well on QA.

## 5.2 Model Type

RoBERTa consistently outperforms BERT, and DeBERTaV3 consistently outperforms RoBERTa

for every language and model size (Table 4). The regression model estimates an 9.1 point improvement for RoBERTa over BERT. DeBERTaV3 shows an additional 24.6 point improvement on RoBERTa, which is nearly double that of RoBERTa on BERT. Due to the absence of Dutch DeBERTaV3 models, we cannot be sure whether monolingual Dutch models would yield this large improvement. It might be that DeBERTaV3 learns a particularly good language-independent representation, which could boost cross-lingual performance more than monolingual performance. English experiments on the GLUE benchmark show the same model performance order, with GLUE scores of 84.1, 88.8 and 91.4 for BERT$_\text{large}$, RoBERTa$_\text{large}$ and DeBERTaV3$_\text{large}$, respectively (He et al., 2023). Regardless of the exact improvement, our findings, and similar results for English (He et al., 2023), suggest that a Dutch DeBERTaV3 would outperform RoBERTa models.

## 5.3 Model Size

The Dutch monolingual language models are only available in *base* model sizes. However, larger models perform better, with an estimated 13.9 point improvement for larger models compared to base-sized models. For XLM-R, the difference is 14.7 points and for the three English model types the differences are 7.7, 14.5 and 17.3 points. This shows that larger models perform better for Dutch regardless of pre-training language.

# 6 Conclusion

We have introduced DUMB, a benchmark with nine Dutch language tasks, including four new tasks and Relative Error Reduction (RER) as a comparative model evaluation metric. The tasks are internally consistent with positive correlations between tasks across models. Some randomness in low-resource task performance might be due to model failure of non-Dutch models.

RobBERT models achieve up to 3.6 RER, but multilingual and even English models can achieve up to 14.4 (XLM-R$_\text{large}$) and 15.7 (DeBERTaV3$_\text{large}$) RER, respectively. Model comparisons across pre-training languages, model types and model sizes reveal that high multilingual and English performance on Dutch tasks can be partially attributed to model size, but to an even larger extent this can be attributed to the DeBERTaV3 model type. A Dutch DeBERTaV3$_\text{large}$

model could achieve a substantially higher estimated RER of 38.0. This estimation shows that there is much room for improving benchmark scores with better future models. Additionally, we encourage evaluating large generative models with DUMB. Our results are based on a robust standard approach and set a strong baseline that can also be useful to evaluate effective prompt engineering.

A public leaderboard is available at dumbench.nl and the benchmark and reference model source code are available at github.com/wietsedv/dumb. The leaderboard will contain all models discussed in this paper, and will be kept up-to-date with newer models and updated versions of the benchmark.

## 7 Limitations

We claim to create a balanced set of tasks, but we do not include each and every NLP task type. Our criterion of balance is about not over-representing specific tasks and task categories, not about completeness. We exclude tasks like parsing and we simplify tasks to be classification tasks, such as WiC for WSD and PR instead of coreference resolution. This standardises model implementations and lets us focus on the PLM instead of task-specific architectures on top of the PLMs.

Secondly, the proposed comparative evaluation metric, Relative Error Reduction, can be considered to not be comparable to aggregate scores of other benchmarks. However, we argue that aggregate scores can never be compared across benchmarks with different tasks and datasets. Moreover, some NLP evaluation metrics such as BLEU are not directly translatable to Relative Error Reduction. This is not a limitation for our benchmark, however, since we only include tasks that have fixed gold labels and therefore use error-based metrics.

Thirdly, our model comparison only contains Transformer-encoder models and no generative language models. Like GLUE, our set of tasks is suitable for fine-tuning encoder models, whereas generative models require prompt design. For this paper and the initial benchmark entries, we aimed to get robust baseline results with a standard fine-tuning approach, a hyper-parameter grid-search and multiple runs. We believe that such robust baseline results are necessary to be able to contextualize highly variable prompt-engineering based approaches.

And finally, our model comparison contains more English than Dutch models. We try to show the effects of several modeling decisions in order to propose a way forward for Dutch modeling, but availability of comparable multilingual and Dutch PLMs is limited. Size and architecture factors may differ for monolingual Dutch models, but we do not currently have the models to make that evaluation. We specifically create this benchmark to motivate further development of Dutch PLMs.

## 8 Ethics Statement

Our proposed benchmark consists of existing datasets and derivatives of existing datasets. These source datasets have been published under various licenses, as is discussed in Section 2. Moreover, we list all dataset sources and applicable licenses in the README of our source code, as well as the leaderboard website. We made sure that all datasets are freely available and permit academic use, though not all permit (modified) redistribution. Therefore, we provide a combination of automated and manual methods for downloading these datasets from official sources. A single script can be used to pre-process the data in a deterministic way to recreate our benchmark.

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

## A  Task examples

This appendix contains example items for each DUMB task, selected from training data.

### POS: Part-of-Speech Tagging (Lassy)

Provide POS tags for every word in the sentence.

| Sentence | Tagged Sentence |
|---|---|
| Scoubidou-touwtjes zijn veilig in de hand, maar niet in de mond. | `[N|soort|mv|dim` Scoubidou-touwtjes] `[WW|pv|tgw|mv` zijn] `[ADJ|vrij|basis|zonder` veilig] `[VZ`init in] `[LID|bep|stan|rest` de] `[N|soort|ev|basis|zijd|stan` hand] `[LET` ,] `[VG|neven` maar] `[BW` niet] `[VZ|init` in] `[LID|bep|stan|rest` de] `[N|soort|ev|basis|zijd|stan` mond] `[LET` .] |

### NER: Named Entity Recognition (SoNaR)

Mark all named entities in the sentence.

| Sentence | Tagged Sentence |
|---|---|
| Topman Jack Welch van het Amerikaanse industriële concern General Electric (GE) verwerpt het aanbod van zijn collega van Honeywell om de beoogde fusie van de twee ondernemingen te redden. | Topman [`PERSON` Jack Welch] van het [`LOCATION` Amerikaanse] industriële concern [`ORGANIZATION` General Electric] ([`ORGANIZATION` GE]) verwerpt het aanbod van zijn collega van [`ORGANIZATION` Honeywell] om de beoogde fusie van de twee ondernemingen te redden. |
| De radar wordt dit weekend gepresenteerd op het Vogelfestival in het natuurgebied de Oostvaardersplassen in Lelystad. | De radar wordt dit weekend gepresenteerd op het [`EVENT` Vogelfestival] in het natuurgebied de [`LOCATION` Oostvaardersplassen] in [`LOCATION` Lelystad]. |

### WSD: Word Sense Disambiguation (WiC-NL)

Determine whether the marked words in each sentence have the same sense.

| Sentence 1 | Sentence 2 | Label |
|---|---|---|
| In bijna elk **mechanisch** apparaat zijn wel assen te vinden. *(mechanical device)* | Mannen daarentegen zijn meer geboeid door mechaniek en willen nagenoeg altijd een **mechanisch** uurwerk. *(mechanical clock)* | same |
| Het merendeel lijkt een **ijzige** kalmte over zich heen te hebben. *(icy calm)* | De schaatsgrootheden uit de tijd van de wollen muts en de **ijzig** koude buitenbanen met storm en sneeuw kunnen worden vergeleken met de schaatsgrootheden uit de tijd van gestoomlijnde pakken, klapschaats en en geconditioneerde binnenbanen. *(icy temperature)* | different |

### PR: Pronoun Resolution (DPR)

Determine whether the marked pronoun refers to the marked expression.

| Text | Label |
|---|---|
| Toen kwam de aanslag op New York en **de generaal**, intussen president, werd voor de keuze gesteld. **Hij** nam het binnenlandse risico: confrontatie met zijn islamitische militanten, in plaats van met de Verenigde Staten. *(the general, He)* | same |
| Di Rupo weet waarom **hij** zich verzet tegen **het privatiseringsbeginsel**. *(he, the privatization principle)* | different |

### CR: Causal Reasoning (COPA-NL)

Choose the most plausible cause or effect, given a premise.

| Premise | Choice 1 | Choice 2 | Label |
|---|---|---|---|
| De vrouw bungelde het koekje boven de hond. *(The woman dangled the biscuit above the dog.)* | De hond sprong op.*(The dog jumped up.)* | De hond krabde aan zijn vacht. *(The dog scratched its fur.)* | Choice 1 |
| De vrouw voelde zich eenzaam. *(The woman felt lonely.)* | Ze renoveerde haar keuken. *(She renovated her kitchen.)* | Ze adopteerde een kat. *(She adopted a cat.)* | Choice 2 |

### NLI: Natural Language Inference (SICK-NL)

Classify whether the first sentence entails or contradicts the second sentence.

| Sentence 1 | Sentence 2 | Label |
|---|---|---|
| Een man springt in een leeg bad *(A man is jumping into an empty pool)* | Een man springt in een vol zwembad *(A man is jumping into a full pool)* | contradiction |
| Een man met een trui is de bal aan het dunken bij een basketbalwedstrijd *(A man with a jersey is dunking the ball at a basketball game)* | De bal wordt gedunkt door een man met een trui bij een basketbalwedstrijd *(The ball is being dunked by a man with a jersey at a basketball game)* | entailment |
| Drie kinderen zitten in de bladeren *Three kids are sitting in the leaves* | Drie kinderen springen in de bladeren *(Three kids are jumping in the leaves)* | neutral |

### ALD: Abusive Language Detection (DALC)

Classify whether the tweet is abusive or offensive.

| Text | Label |
|---|---|
| Ach @USER wie neemt die nog serieus. Het gezellige dikkertje dat propogandeerde dat dik zijn, wat extra vet niet erg is en dat we gewoon lekker ongezond moeten eten wanneer we dat willen. En nu klagen over de kwetsbaren wat juist diegene zijn met teveel vetcellen. *(fat shaming)* | abusive |
| @USER OMDAT VROUWEN MOEILIJKE WEZENS ZIJN *(misogynistic)* | offensive |

**SA: Sentiment Analysis (DBRD)**

Classify whether the review is positive or negative.

| Text | Label |
|---|---|
| Het verhaal speelt zich af aan het einde van de 19e eeuw. Boeiend van begin tot eind, geeft het een inkijkje in het leven van arbeiders en wetenschappers in Barcelona. De industriële revolutie is net begonnen en de effecten daarvan tekenen zich af. Grote veranderingen op het gebied van de medische wetenschap spelen op de achtergrond van liefde, vriendschap, betrokkenheid en verraad. Fictie wordt vermengd met historische feiten op een meeslepende manier, pakkend begin, verrassend einde. Aanrader! *(explicit recommendation)* | positive |
| Eerlijk gezegd vindt ik dat dit boek vreemd is geschreven. De verhaallijnen gaan door elkaar heen en dat maakt het heel onduidelijk. Het onderwerp is wel goed bedacht *(odd writing style and entangled story lines)* | negative |

**QA: Question Answering (SQuAD-NL)**

Locate the answer to a question in a given paragraph, or classify the question as unanswerable.

| Question | Context and Answer |
|---|---|
| Wat is Saksische tuin in het Pools? | Vlakbij, in **Ogród Saski** (de Saksische Tuin), was het Zomertheater in gebruik van 1870 tot 1939, en in het interbellum omvatte het theatercomplex ook Momus, het eerste literaire cabaret van Warschau, en Melodram, het muziektheater van Leon Schiller. Het Wojciech Bogusławski Theater (1922-26) was het beste voorbeeld van "Pools monumentaal theater". Vanaf het midden van de jaren dertig huisvestte het Great Theatre-gebouw het Upati Institute of Dramatic Arts - de eerste door de staat gerunde academie voor dramatische kunst, met een acteerafdeling en een regie-afdeling. |

## B  Training setup and hyper-parameters

Our main results are based on 14 pre-trained language models, 9 tasks and 5 test runs per model and task, totalling 630 runs. However, the total number of runs is much larger due to experimentation and hyper-parameter search. The total amount of runs within the hyper-parameter grids (can be found on the next two pages) is 7,504. The hyper-parameter search ranges can be found on the next two pages. Training durations per vary from seconds to hours, depending on task, model size and training epochs.

All models were trained on single A100 (40GB) GPUs with the implementations on `github.com/wietsedv/dumb/tree/main/trainers` and the following hyper-parameters:

- Batch Size: 32
- Weight Decay: 0
- Learning Rate Decay: Linear
- Optimizer: Adam
- Adam $\beta_1$: 0.9
- Adam $\beta_2$: 0.999
- Adam $\epsilon$: 1e-8
- Gradient Clipping: 1.0
- Epochs: *see table*
- Dropout: *see table*
- Learning Rate: *see table*
- Warmup: *see table*

Roughly estimated training durations across models and hyper-parameters are:

- POS: 30 minutes
- NER: 30 minutes
- WSD: 8 minutes
- PR: 2 minutes
- CR: 2 minutes
- NLI: 8 minutes
- SA: 45 minutes
- ALD: 10 minutes
- QA: 4 hours

Combined grid search and test run times add up to 105 GPU hours per pre-trained model and 61 days for the 14 models combined. Additional experiments that did not end up in the paper may add about 50% extra GPU hours. When evaluating new models, we advice experimenting with smaller grids based on our optimal parameters.

# C   Model Performance Regression Model

This table shows the linear regression model discussed in Section 5.

```
                          OLS Regression Results
==============================================================================
Dep. Variable:                    RER   R-squared:                       0.836
Model:                            OLS   Adj. R-squared:                  0.733
Method:                 Least Squares   F-statistic:                     8.127
Date:                Tue, 13 Jun 2023   Prob (F-statistic):            0.00535
Time:                        18:34:39   Log-Likelihood:                -47.201
No. Observations:                  14   AIC:                             106.4
Df Residuals:                       8   BIC:                             110.2
Df Model:                           5
Covariance Type:            nonrobust
==============================================================================
                            coef    std err          t      P>|t|      [0.025      0.975]
------------------------------------------------------------------------------
Intercept                 -9.5663      6.507     -1.470      0.180     -24.571       5.438
Language[T.english]      -28.8601      7.480     -3.858      0.005     -46.109     -11.611
Language[T.multilingual]  -1.5577      7.072     -0.220      0.831     -17.865      14.750
Model[T.debertav3]        33.6502      7.279      4.623      0.002      16.865      50.435
Model[T.roberta]           9.0766      6.053      1.499      0.172      -4.882      23.035
Size[T.large]             13.8761      6.333      2.191      0.060      -0.728      28.480
==============================================================================
Omnibus:                        3.330   Durbin-Watson:                   2.501
Prob(Omnibus):                  0.189   Jarque-Bera (JB):                2.099
Skew:                          -0.941   Prob(JB):                        0.350
Kurtosis:                       2.768   Cond. No.                         5.86
==============================================================================
```

| Task | Model | Epochs | Warmup | LR | Dropout | Dev | Test |
|------|-------|--------|--------|----|---------|-----|------|
| POS | BERTje | {1, **3**, 5} | {0.0, **0.3**} | {3e-05, 5e-05, **0.0001**} | {0.0, **0.1**} | 98.4 | 97.8 |
| POS | RobBERT$_{V1}$ | {1, 3, **5**} | {0.0, **0.3**} | {3e-05, 5e-05, **0.0001**} | {0.0, **0.1**} | 98.6 | 98.1 |
| POS | RobBERT$_{V2}$ | {1, **3**, 5} | {0.0, **0.3**} | {3e-05, 5e-05, **0.0001**} | {0.0, **0.1**} | 98.7 | 98.2 |
| POS | RobBERT$_{2022}$ | {1, 3, **5**} | {**0.0**, 0.3} | {3e-05, **5e-05**, 0.0001} | {**0.0**, 0.1} | 98.6 | 98.2 |
| POS | mBERT$_{cased}$ | {1, **5**} | {0.0, **0.3**} | {3e-05, **5e-05**, 0.0001} | {0.0, **0.1**} | 98.6 | 97.9 |
| POS | XLM-R$_{base}$ | {1, **3**, 5} | {**0.0**, 0.3} | {3e-05, 5e-05, **0.0001**} | {0.0, **0.1**} | 98.7 | 98.1 |
| POS | mDeBERTaV3$_{base}$ | {1, **3**, 5} | {0.0, **0.3**} | {3e-05, 5e-05, **0.0001**} | {0.0, **0.1**} | 98.7 | 98.2 |
| POS | XLM-R$_{large}$ | {1, **3**, 5} | {0.0, **0.3**} | {3e-05, **5e-05**, 0.0001} | {**0.0**, 0.1} | 98.8 | 98.4 |
| POS | BERT$_{base}$ | {1, 3, **5**} | {**0.0**, 0.3} | {3e-05, **5e-05**, 0.0001} | {**0.0**, 0.1} | 97.9 | 97.3 |
| POS | RoBERTa$_{base}$ | {1, 3, **5**} | {0.0, **0.3**} | {3e-05, 5e-05, **0.0001**} | {0.0, **0.1**} | 98.2 | 97.6 |
| POS | DeBERTaV3$_{base}$ | {1, 3, **5**} | {0.0, **0.3**} | {3e-05, 5e-05, **0.0001**} | {0.0, **0.1**} | 98.4 | 97.9 |
| POS | BERT$_{large}$ | {1, 3, **5**} | {0.0, **0.3**} | {3e-05, **5e-05**, 0.0001} | {**0.0**, 0.1} | 98.2 | 97.5 |
| POS | RoBERTa$_{large}$ | {1, 3, **5**} | {**0.0**, 0.3} | {3e-05, **5e-05**, 0.0001} | {**0.0**, 0.1} | 98.5 | 97.9 |
| POS | DeBERTaV3$_{large}$ | {1, 3, **5**} | {0.0, **0.3**} | {**3e-05**, 5e-05, 0.0001} | {0.0, **0.1**} | 98.6 | 98.2 |
| NER | BERTje | {1, 3, **5**} | {**0.0**, 0.3} | {1e-05, 3e-05, **5e-05**} | {0.0, **0.1**} | 85.1 | 86.1 |
| NER | RobBERT$_{V1}$ | {1, 3, **5**} | {0.0, **0.3**} | {1e-05, 3e-05, **5e-05**} | {0.0, **0.1**} | 84.2 | 83.5 |
| NER | RobBERT$_{V2}$ | {1, 3, **5**} | {**0.0**, 0.3} | {1e-05, **3e-05**, 5e-05} | {**0.0**, 0.1} | 85.5 | 86.7 |
| NER | RobBERT$_{2022}$ | {1, 3, **5**} | {**0.0**, 0.3} | {1e-05, 3e-05, **5e-05**} | {0.0, **0.1**} | 85.7 | 87.2 |
| NER | mBERT$_{cased}$ | {1, 3, **5**} | {0.0, **0.3**} | {1e-05, 3e-05, **5e-05**} | {0.0, **0.1**} | 86.6 | 87.4 |
| NER | XLM-R$_{base}$ | {1, 3, **5**} | {**0.0**, 0.3} | {1e-05, **3e-05**, 5e-05} | {0.0, **0.1**} | 85.9 | 87.6 |
| NER | mDeBERTaV3$_{base}$ | {1, 3, **5**} | {**0.0**, 0.3} | {1e-05, 3e-05, **5e-05**} | {0.0, **0.1**} | 85.1 | 88.5 |
| NER | XLM-R$_{large}$ | {1, 3, **5**} | {0.0, **0.3**} | {**1e-05**, 3e-05, 5e-05} | {0.0, **0.1**} | 87.5 | 90.3 |
| NER | BERT$_{base}$ | {1, 3, **5**} | {**0.0**, 0.3} | {1e-05, 3e-05, **5e-05**} | {0.0, **0.1**} | 81.7 | 81.9 |
| NER | RoBERTa$_{base}$ | {1, **3**, 5} | {**0.0**, 0.3} | {1e-05, **3e-05**, 5e-05} | {**0.0**, 0.1} | 83.7 | 82.3 |
| NER | DeBERTaV3$_{base}$ | {1, 3, **5**} | {**0.0**, 0.3} | {1e-05, 3e-05, **5e-05**} | {0.0, **0.1**} | 83.6 | 86.4 |
| NER | BERT$_{large}$ | {1, **3**, 5} | {**0.0**, 0.3} | {1e-05, 3e-05, **5e-05**} | {0.0, **0.1**} | 83.3 | 82.5 |
| NER | RoBERTa$_{large}$ | {1, 3, **5**} | {0.0, **0.3**} | {**1e-05**, 3e-05, 5e-05} | {0.0, **0.1**} | 83.8 | 84.4 |
| NER | DeBERTaV3$_{large}$ | {1, 3, **5**} | {0.0, **0.3**} | {**1e-05**, 3e-05, 5e-05} | {**0.0**, 0.1} | 88.2 | 87.6 |
| WSD | BERTje | {**1**, 3, 5, 10} | {0.0, **0.3**} | {1e-05, 3e-05, 5e-05, **0.0001**} | {**0.0**, 0.1} | 67.9 | 65.9 |
| WSD | RobBERT$_{V1}$ | {1, **3**, 5, 10} | {**0.0**, 0.3} | {1e-05, 3e-05, **5e-05**, 0.0001} | {0.0, **0.1**} | 61.3 | 60.6 |
| WSD | RobBERT$_{V2}$ | {1, 3, 5, **10**} | {**0.0**, 0.3} | {1e-05, **3e-05**, 5e-05, 0.0001} | {0.0, **0.1**} | 66.3 | 64.1 |
| WSD | RobBERT$_{2022}$ | {1, **3**, 5, 10} | {**0.0**, 0.3} | {1e-05, 3e-05, 5e-05, **0.0001**} | {0.0, **0.1**} | 67.0 | 63.7 |
| WSD | mBERT$_{cased}$ | {1, **3**, 5, 10} | {**0.0**, 0.3} | {**1e-05**, 3e-05, 5e-05, 0.0001} | {0.0, **0.1**} | 68.2 | 68.5 |
| WSD | XLM-R$_{base}$ | {1, 3, **5**, 10} | {0.0, **0.3**} | {1e-05, **3e-05**, 5e-05, 0.0001} | {0.0, **0.1**} | 66.7 | 66.5 |
| WSD | mDeBERTaV3$_{base}$ | {1, **3**, 5, 10} | {**0.0**, 0.3} | {1e-05, 3e-05, 5e-05, **0.0001**} | {0.0, **0.1**} | 69.8 | 69.6 |
| WSD | XLM-R$_{large}$ | {1, 3, 5, **10**} | {0.0, **0.3**} | {**1e-05**, 3e-05, 5e-05, 0.0001} | {**0.0**, 0.1} | 73.0 | 73.1 |
| WSD | BERT$_{base}$ | {1, **3**, 5, 10} | {**0.0**, 0.3} | {**1e-05**, 3e-05, 5e-05, 0.0001} | {0.0, **0.1**} | 60.0 | 58.2 |
| WSD | RoBERTa$_{base}$ | {1, **3**, 5, 10} | {**0.0**, 0.3} | {1e-05, **3e-05**, 5e-05, 0.0001} | {**0.0**, 0.1} | 62.6 | 61.1 |
| WSD | DeBERTaV3$_{base}$ | {1, 3, **5**, 10} | {**0.0**, 0.3} | {1e-05, **3e-05**, 5e-05, 0.0001} | {0.0, **0.1**} | 66.4 | 64.4 |
| WSD | BERT$_{large}$ | {1, 3, 5, **10**} | {0.0, **0.3**} | {**1e-05**, 3e-05, 5e-05, 0.0001} | {0.0, **0.1**} | 60.2 | 57.2 |
| WSD | RoBERTa$_{large}$ | {1, 3, 5, **10**} | {**0.0**, 0.3} | {**1e-05**, 3e-05, 5e-05, 0.0001} | {0.0, **0.1**} | 64.3 | 59.1 |
| WSD | DeBERTaV3$_{large}$ | {1, 3, 5, 10} | {0.0, **0.3**} | {1e-05, **3e-05**, 5e-05, 0.0001} | {0.0, **0.1**} | 71.3 | 70.2 |
| PR | BERTje | {1, 3, 5, 10, **20**} | {**0.0**, 0.3} | {1e-05, 3e-05, 5e-05, **0.0001**} | {0.0, **0.1**} | 69.7 | 65.8 |
| PR | RobBERT$_{V1}$ | {1, 3, 5, **10**, 20} | {**0.0**, 0.3} | {1e-05, 3e-05, 5e-05, **0.0001**} | {**0.0**, 0.1} | 66.9 | 57.3 |
| PR | RobBERT$_{V2}$ | {1, 3, 5, 10, **20**} | {**0.0**, 0.3} | {1e-05, 3e-05, 5e-05, **0.0001**} | {0.0, **0.1**} | 69.0 | 65.8 |
| PR | RobBERT$_{2022}$ | {1, 3, **5**, 10, 20} | {**0.0**, 0.3} | {1e-05, 3e-05, 5e-05, **0.0001**} | {**0.0**, 0.1} | 71.1 | 65.2 |
| PR | mBERT$_{cased}$ | {1, 3, 5, 10, **20**} | {**0.0**, 0.3} | {1e-05, **3e-05**, 5e-05, 0.0001} | {0.0, **0.1**} | 70.4 | 62.0 |
| PR | XLM-R$_{base}$ | {1, 3, **5**, 10, 20} | {**0.0**, 0.3} | {1e-05, 3e-05, 5e-05, **0.0001**} | {**0.0**, 0.1} | 64.1 | 57.4 |
| PR | mDeBERTaV3$_{base}$ | {1, 3, **5**, 10, 20} | {**0.0**, 0.3} | {1e-05, 3e-05, 5e-05, **0.0001**} | {0.0, **0.1**} | 69.7 | 58.7 |
| PR | XLM-R$_{large}$ | {1, 3, 5, 10, **20**} | {**0.0**, 0.3} | {1e-05, **3e-05**, 5e-05, 0.0001} | {**0.0**, 0.1} | 64.8 | 60.4 |
| PR | BERT$_{base}$ | {1, 3, 5, **10**, 20} | {**0.0**, 0.3} | {1e-05, 3e-05, **5e-05**, 0.0001} | {**0.0**, 0.1} | 65.5 | 57.6 |
| PR | RoBERTa$_{base}$ | {1, 3, **5**, 10, 20} | {0.0, **0.3**} | {1e-05, 3e-05, **5e-05**, 0.0001} | {**0.0**, 0.1} | 68.3 | 60.4 |
| PR | DeBERTaV3$_{base}$ | {1, **3**, 5, 10, 20} | {0.0, **0.3**} | {1e-05, 3e-05, **5e-05**, 0.0001} | {**0.0**, 0.1} | 69.0 | 57.1 |
| PR | BERT$_{large}$ | {**1**, 3, 5, 10, 20} | {0.0, **0.3**} | {1e-05, 3e-05, **5e-05**, 0.0001} | {**0.0**, 0.1} | 64.1 | 55.8 |
| PR | RoBERTa$_{large}$ | {1, 3, 5, 10, **20**} | {0.0, **0.3**} | {**1e-05**, 3e-05, 5e-05, 0.0001} | {**0.0**, 0.1} | 61.3 | 56.6 |
| PR | DeBERTaV3$_{large}$ | {1, 3, 5, 10, **20**} | {**0.0**, 0.3} | {1e-05, 3e-05, **5e-05**, 0.0001} | {0.0, **0.1**} | 71.8 | 60.9 |

| Task | Model | Epochs | Warmup | LR | Dropout | Dev | Test |
|------|-------|--------|--------|----|---------|-----|------|
| CR | BERTje | {**1**, 3, 5, 10, 20} | {0.0, **0.3**} | {1e-05, 3e-05, **5e-05**, 0.0001} | {**0.0**, 0.1} | 71.0 | 62.0 |
| CR | RobBERT$_{V1}$ | {1, 3, 5, **10**, 20} | {0.0, **0.3**} | {1e-05, **3e-05**, 5e-05, 0.0001} | {0.0, **0.1**} | 69.0 | 56.4 |
| CR | RobBERT$_{V2}$ | {1, 3, 5, **10**, 20} | {0.0, **0.3**} | {1e-05, 3e-05, **5e-05**, 0.0001} | {0.0, **0.1**} | 68.0 | 56.2 |
| CR | RobBERT$_{2022}$ | {1, 3, 5, **10**, 20} | {0.0, **0.3**} | {1e-05, **3e-05**, 5e-05, 0.0001} | {0.0, **0.1**} | 71.0 | 55.4 |
| CR | mBERT$_{cased}$ | {1, 3, 5, **10**, 20} | {**0.0**, 0.3} | {1e-05, 3e-05, **5e-05**, 0.0001} | {0.0, **0.1**} | 61.0 | 55.0 |
| CR | XLM-R$_{base}$ | {1, 3, **5**, 10, 20} | {**0.0**, 0.3} | {1e-05, **3e-05**, 5e-05, 0.0001} | {0.0, **0.1**} | 70.0 | 51.8 |
| CR | mDeBERTaV3$_{base}$ | {1, **3**, 5, 10, 20} | {0.0, **0.3**} | {1e-05, 3e-05, 5e-05, **0.0001**} | {0.0, **0.1**} | 81.0 | 69.5 |
| CR | XLM-R$_{large}$ | {1, 3, 5, 10, **20**} | {**0.0**, 0.3} | {1e-05, **3e-05**, 5e-05, 0.0001} | {**0.0**, 0.1} | 66.0 | 52.2 |
| CR | BERT$_{base}$ | {1, 3, **5**, 10, 20} | {0.0, **0.3**} | {1e-05, **3e-05**, 5e-05, 0.0001} | {0.0, **0.1**} | 65.0 | 51.2 |
| CR | RoBERTa$_{base}$ | {1, 3, **5**, 10, 20} | {0.0, **0.3**} | {1e-05, **3e-05**, 5e-05, 0.0001} | {0.0, **0.1**} | 70.0 | 52.8 |
| CR | DeBERTaV3$_{base}$ | {1, 3, 5, **10**, 20} | {0.0, **0.3**} | {1e-05, 3e-05, 5e-05, **0.0001**} | {0.0, **0.1**} | 61.0 | 51.4 |
| CR | BERT$_{large}$ | {1, 3, 5, **10**, 20} | {**0.0**, 0.3} | {1e-05, 3e-05, **5e-05**, 0.0001} | {**0.0**, 0.1} | 64.0 | 50.2 |
| CR | RoBERTa$_{large}$ | {1, **3**, 5, 10, 20} | {0.0, **0.3**} | {**1e-05**, 3e-05, 5e-05, 0.0001} | {**0.0**, 0.1} | 66.0 | 52.1 |
| CR | DeBERTaV3$_{large}$ | {1, 3, 5, 10, **20**} | {0.0, **0.3**} | {**1e-05**, 3e-05, 5e-05, 0.0001} | {**0.0**, 0.1} | 82.0 | 75.4 |
| NLI | BERTje | {1, **3**, 5, 10} | {0.0, **0.3**} | {1e-05, 3e-05, 5e-05, **0.0001**} | {**0.0**, 0.1} | 84.2 | 85.2 |
| NLI | RobBERT$_{V1}$ | {1, **3**, 5, 10} | {0.0, **0.3**} | {1e-05, 3e-05, **5e-05**, 0.0001} | {**0.0**, 0.1} | 83.2 | 83.3 |
| NLI | RobBERT$_{V2}$ | {1, 3, 5, **10**} | {0.0, **0.3**} | {1e-05, 3e-05, **5e-05**, 0.0001} | {0.0, **0.1**} | 85.5 | 84.6 |
| NLI | RobBERT$_{2022}$ | {1, 3, 5, **10**} | {**0.0**, 0.3} | {1e-05, 3e-05, **5e-05**, 0.0001} | {**0.0**, 0.1} | 85.1 | 85.6 |
| NLI | mBERT$_{cased}$ | {1, 3, **5**, 10} | {**0.0**, 0.3} | {1e-05, 3e-05, **5e-05**, 0.0001} | {**0.0**, 0.1} | 83.8 | 84.3 |
| NLI | XLM-R$_{base}$ | {1, 3, **5**, 10} | {0.0, **0.3**} | {1e-05, 3e-05, **5e-05**, 0.0001} | {**0.0**, 0.1} | 85.9 | 85.5 |
| NLI | mDeBERTaV3$_{base}$ | {1, **3**, 5, 10} | {0.0, **0.3**} | {1e-05, 3e-05, **5e-05**, 0.0001} | {0.0, **0.1**} | 89.1 | 88.9 |
| NLI | XLM-R$_{large}$ | {1, 3, **5**, 10} | {0.0, **0.3**} | {1e-05, **3e-05**, 5e-05, 0.0001} | {0.0, **0.1**} | 89.1 | 88.8 |
| NLI | BERT$_{base}$ | {1, 3, 5, **10**} | {**0.0**, 0.3} | {1e-05, **3e-05**, 5e-05, 0.0001} | {0.0, **0.1**} | 81.4 | 82.3 |
| NLI | RoBERTa$_{base}$ | {1, 3, **5**, 10} | {**0.0**, 0.3} | {1e-05, 3e-05, 5e-05, **0.0001**} | {**0.0**, 0.1} | 83.2 | 82.3 |
| NLI | DeBERTaV3$_{base}$ | {1, 3, **5**, 10} | {**0.0**, 0.3} | {1e-05, 3e-05, **5e-05**, 0.0001} | {0.0, **0.1**} | 87.1 | 86.5 |
| NLI | BERT$_{large}$ | {1, 3, **5**, 10} | {**0.0**, 0.3} | {1e-05, **3e-05**, 5e-05, 0.0001} | {**0.0**, 0.1} | 83.4 | 82.9 |
| NLI | RoBERTa$_{large}$ | {1, 3, 5, **10**} | {**0.0**, 0.3} | {**1e-05**, 3e-05, 5e-05, 0.0001} | {**0.0**, 0.1} | 84.2 | 83.9 |
| NLI | DeBERTaV3$_{large}$ | {1, **3**, 5, 10} | {**0.0**, 0.3} | {**1e-05**, 3e-05, 5e-05, 0.0001} | {**0.0**, 0.1} | 88.9 | 89.1 |
| SA | BERTje | {1, **2**, 3} | {0.0, **0.3**} | {**3e-05**, 5e-05, 0.0001} | {0.0, **0.1**} | 94.8 | 93.3 |
| SA | RobBERT$_{V1}$ | {1, 2, **3**} | {0.0, **0.3**} | {3e-05, 5e-05, **0.0001**} | {**0.0**, 0.1} | 91.2 | 89.4 |
| SA | RobBERT$_{V2}$ | {1, 2, **3**} | {0.0, **0.3**} | {3e-05, **5e-05**, 0.0001} | {**0.0**, 0.1} | 95.4 | 93.2 |
| SA | RobBERT$_{2022}$ | {1, 2, **3**} | {0.0, **0.3**} | {**3e-05**, 5e-05, 0.0001} | {**0.0**, 0.1} | 95.2 | 93.5 |
| SA | mBERT$_{cased}$ | {1, **2**, 3} | {0.0, **0.3**} | {3e-05, **5e-05**, 0.0001} | {**0.0**, 0.1} | 92.8 | 90.5 |
| SA | XLM-R$_{base}$ | {1, **2**, 3} | {0.0, **0.3**} | {**3e-05**, 5e-05, 0.0001} | {**0.0**, 0.1} | 95.6 | 93.0 |
| SA | mDeBERTaV3$_{base}$ | {1, **2**, 3} | {0.0, **0.3**} | {**3e-05**, 5e-05, 0.0001} | {**0.0**, 0.1} | 94.2 | 93.5 |
| SA | XLM-R$_{large}$ | {1, **2**, 3} | {0.0, **0.3**} | {**3e-05**, 5e-05, 0.0001} | {**0.0**, 0.1} | 96.4 | 94.2 |
| SA | BERT$_{base}$ | {1, **2**, 3} | {**0.0**, 0.3} | {3e-05, **5e-05**, 0.0001} | {**0.0**, 0.1} | 81.0 | 79.6 |
| SA | RoBERTa$_{base}$ | {1, 2, **3**} | {**0.0**, 0.3} | {**3e-05**, 5e-05, 0.0001} | {**0.0**, 0.1} | 86.8 | 86.6 |
| SA | DeBERTaV3$_{base}$ | {1, 2, **3**} | {0.0, **0.3**} | {**3e-05**, 5e-05, 0.0001} | {0.0, **0.1**} | 92.4 | 92.3 |
| SA | BERT$_{large}$ | {1, 2, **3**} | {0.0, **0.3**} | {**3e-05**, 5e-05, 0.0001} | {**0.0**, 0.1} | 82.4 | 82.6 |
| SA | RoBERTa$_{large}$ | {1, 2, **3**} | {0.0, **0.3**} | {**3e-05**, 5e-05, 0.0001} | {**0.0**, 0.1} | 87.4 | 89.0 |
| SA | DeBERTaV3$_{large}$ | {**1**, 2, 3} | {0.0, **0.3**} | {**3e-05**, 5e-05, 0.0001} | {0.0, **0.1**} | 93.6 | 92.8 |
| ALD | BERTje | {1, 3, **5**, 10} | {0.0, **0.3**} | {3e-05, **5e-05**, 0.0001} | {0.0, **0.1**} | 67.4 | 58.8 |
| ALD | RobBERT$_{V1}$ | {1, 3, **5**, 10} | {**0.0**, 0.3} | {3e-05, **5e-05**, 0.0001} | {0.0, **0.1**} | 66.3 | 60.8 |
| ALD | RobBERT$_{V2}$ | {1, 3, **5**, 10} | {0.0, **0.3**} | {**3e-05**, 5e-05, 0.0001} | {0.0, **0.1**} | 70.1 | 63.7 |
| ALD | RobBERT$_{2022}$ | {1, 3, **5**, 10} | {0.0, **0.3**} | {**3e-05**, 5e-05, 0.0001} | {**0.0**, 0.1} | 67.9 | 66.6 |
| ALD | mBERT$_{cased}$ | {1, 3, 5, **10**} | {0.0, **0.3**} | {3e-05, 5e-05, **0.0001**} | {0.0, **0.1**} | 64.0 | 56.9 |
| ALD | XLM-R$_{base}$ | {1, 3, 5, **10**} | {0.0, **0.3**} | {**3e-05**, 5e-05, 0.0001} | {0.0, **0.1**} | 65.8 | 60.2 |
| ALD | mDeBERTaV3$_{base}$ | {1, 3, 5, **10**} | {0.0, **0.3**} | {3e-05, **5e-05**, 0.0001} | {0.0, **0.1**} | 68.0 | 63.9 |
| ALD | XLM-R$_{large}$ | {1, 3, **5**, 10} | {0.0, **0.3**} | {**3e-05**, 5e-05, 0.0001} | {**0.0**, 0.1} | 70.3 | 66.6 |
| ALD | BERT$_{base}$ | {1, 3, 5, **10**} | {0.0, **0.3**} | {3e-05, 5e-05, **0.0001**} | {**0.0**, 0.1} | 60.1 | 52.2 |
| ALD | RoBERTa$_{base}$ | {1, 3, **5**, 10} | {**0.0**, 0.3} | {**3e-05**, 5e-05, 0.0001} | {**0.0**, 0.1} | 63.3 | 52.2 |
| ALD | DeBERTaV3$_{base}$ | {1, **3**, 5, 10} | {0.0, **0.3**} | {**3e-05**, 5e-05, 0.0001} | {**0.0**, 0.1} | 64.9 | 60.2 |
| ALD | BERT$_{large}$ | {1, 3, 5, **10**} | {0.0, **0.3**} | {**3e-05**, 5e-05, 0.0001} | {0.0, **0.1**} | 62.3 | 55.6 |
| ALD | RoBERTa$_{large}$ | {1, 3, 5, **10**} | {0.0, **0.3**} | {**3e-05**, 5e-05, 0.0001} | {**0.0**, 0.1} | 64.0 | 59.3 |
| ALD | DeBERTaV3$_{large}$ | {1, 3, 5, **10**} | {0.0, **0.3**} | {**3e-05**, 5e-05, 0.0001} | {0.0, **0.1**} | 68.1 | 64.0 |

| Task | Model | Epochs | Warmup | LR | Dropout | Dev | Test |
|------|-------|--------|--------|-----|---------|-----|------|
| QA | BERTje | {**2**} | {0.0, **0.3**} | {3e-05, **5e-05**, 0.0001} | {0.0, **0.1**} | 70.3 | 70.3 |
| QA | RobBERT$_{V1}$ | {**2**} | {**0.0**, 0.3} | {3e-05, 5e-05, **0.0001**} | {0.0, **0.1**} | 64.2 | 64.6 |
| QA | RobBERT$_{V2}$ | {**2**} | {0.0, **0.3**} | {3e-05, 5e-05, **0.0001**} | {**0.0**, 0.1} | 68.8 | 71.0 |
| QA | RobBERT$_{2022}$ | {**2**} | {**0.0**, 0.3} | {3e-05, 5e-05, **0.0001**} | {**0.0**, 0.1} | 69.0 | 70.3 |
| QA | mBERT$_{cased}$ | {**2**} | {0.0, **0.3**} | {3e-05, **5e-05**, 0.0001} | {0.0, **0.1**} | 71.7 | 72.4 |
| QA | XLM-R$_{base}$ | {**2**} | {**0.0**, 0.3} | {3e-05, **5e-05**, 0.0001} | {**0.0**, 0.1} | 73.8 | 74.0 |
| QA | mDeBERTaV3$_{base}$ | {**2**} | {**0.0**, 0.3} | {3e-05, **5e-05**, 0.0001} | {0.0, **0.1**} | 79.9 | 79.0 |
| QA | XLM-R$_{large}$ | {**2**} | {**0.0**, 0.3} | {**3e-05**, 5e-05, 0.0001} | {0.0, **0.1**} | 82.6 | 81.4 |
| QA | BERT$_{base}$ | {**2**} | {0.0, **0.3**} | {3e-05, **5e-05**, 0.0001} | {0.0, **0.1**} | 61.8 | 62.5 |
| QA | RoBERTa$_{base}$ | {**2**} | {0.0, **0.3**} | {3e-05, **5e-05**, 0.0001} | {**0.0**, 0.1} | 68.4 | 69.7 |
| QA | DeBERTaV3$_{base}$ | {**2**} | {**0.0**, 0.3} | {3e-05, **5e-05**, 0.0001} | {**0.0**, 0.1} | 77.7 | 79.1 |
| QA | BERT$_{large}$ | {**2**} | {**0.0**, 0.3} | {**3e-05**, 5e-05, 0.0001} | {**0.0**, 0.1} | 67.0 | 67.2 |
| QA | RoBERTa$_{large}$ | {**2**} | {**0.0**, 0.3} | {**3e-05**, 5e-05, 0.0001} | {**0.0**, 0.1} | 75.0 | 76.2 |
| QA | DeBERTaV3$_{large}$ | {**2**} | {**0.0**, 0.3} | {**3e-05**, 5e-05, 0.0001} | {0.0, **0.1**} | 84.2 | 84.7 |