# OpenReview forum: "DUMB: A Dutch Model Benchmark"
_EMNLP/2023/Conference — EMNLP 2023 Main_

### Official Review · Reviewer_i3Ck · 2023-08-03

**Typos Grammar Style And Presentation Improvements:** 1. When describing the tasks, it is u…
**Soundness:** 4

**Excitement:**

4: Strong: This paper deepens the understanding of some phenomenon or lowers the barriers to an existing research direction.

**Paper Topic And Main Contributions:**

The paper presents a new benchmark for evaluating Dutch models on a set of 9 NLP tasks (4 newly created). These include low, medium and high-resourced tasks and are curated in a way that maximizes overlap within splits across the different tasks. Translated datasets (CoPA, Squad) are checked for accuracy. A total of 14 models of varying size and pretraining techniques are evaluated using a proposed metric: Relative Error Reduction as compared to a baseline, BERTje.

**Questions For The Authors:**

L376-379: Do the monolingual model transfer well to all non-English languages or are there properties of Dutch that should impact the corss-lingual transfer?

Table 4: Does it matter which missing values are estimated? What is the variance on predicting the RER score across different splits in a leave-on-out fashion?

**Reasons To Accept:**

1. The paper presents a thorough comparison of monolingual and multilingual Dutch pre-trained and non-Dutch pre-trained models across 9 different tasks.
2. The paper presents a new metric for performance comparison across different tasks.
3. An analysis based on current results is presented to support the nature of pre-training that could improve performance on this task for future work.

**Reasons To Reject:**

1. L535-536: It is unclear why RER correlation is a more intuitive way to compare performance across tasks and what it actually measures (missing an intuitive explanation rather than just comparing the numbers). Also to clarify, it is clear what RER captures but not what the correlation does.

2. A regression analysis is presented to identify and reason why non-dutch models outperform dutch-pre-trained models, however while the analysis presents some insights on the nature of pretraining and model size, it is unclear why and how the choice of pretraining language would play a role.

**Reproducibility:**

4: Could mostly reproduce the results, but there may be some variation because of sample variance or minor variations in their interpretation of the protocol or method.

**Reviewer Confidence:**

4: Quite sure. I tried to check the important points carefully. It's unlikely, though conceivable, that I missed something that should affect my ratings.

---

> ### Author Rebuttal · Authors · 2023-08-29
>
> We want to thank you for recognizing the value of our work for future research. Below, we will try to respond to your reasons-to-reject (R#) and questions (Q#).
>
> R1; You indicate that we have not motivated our correlation comparison well enough. We use RER for evaluating models, and correlations across models to assess internal consistency of our benchmark tasks. We include a varied set of NLP tasks because different tasks require different aspects of language understanding, and different models may have different strengths and weaknesses. Tasks should not correlate perfectly with each other, but they *should* correlate positively since models that represent Dutch well should perform well on multiple tasks. We think that these insights are important, because a task that is not meaningfully solvable by a LM would get relatively random (non-correlating) scores compared to other tasks. If this is the case for multiple tasks and if only a single aggregate number is used to denote each model’s performance (as is often done when using a benchmark), this will not be observed. We will use some of the extra space in the camera-ready version of the paper to explain the purpose of this sanity check.
>
> R2; you question the influence of pre-training language on performance. Intuitively, LMs that have had more exposure to the Dutch language should perform better on Dutch tasks. We can describe this more explicitly. As you mention, Table 4 shows estimates of model performance for non-existing Dutch and multilingual models, and in Section 5.1, we describe the effects of pre-training language. We at least clearly quantify that multilingual models (that include Dutch) vastly outperform equivalent English models. But English models perform surprisingly well, which is likely influenced by [language contamination](https://aclanthology.org/2022.emnlp-main.233/) and language similarity.
>
> Q1; In the paragraph you cite, we cite previous studies that do not all take language similarity into account when assessing cross-lingual performance. However, intuitively lexical and syntactic similarity would enable higher performance. Unlike most languages of the world, Dutch is closely related to English, as they are both West-Germanic languages. Similar language families, writing systems, and other factors have been demonstrated to increase cross-lingual performance (e.g. [de Vries et al., 2022](https://aclanthology.org/2022.acl-long.529/)). Therefore, the English model results are probably better for Dutch than they would be for most other languages.
>
> Q2; We agree that the current table does not reflect the uncertainty of the estimated values. The statistical model (listed in Appendix C) on which the estimated (gray) values are based, already enables the calculation of variance. Appendix C shows standard errors per predictor, but we will add standard errors per estimation to Table 2. We are aware that standard errors are high (in part due to the presence of only 14 data points), and we will emphasize this uncertainty in the caption and/or main text. The standard errors per estimation are:
>
> | Language | Model | Size | Est. Avg. RER | Std. Err. |
> | -------- | --------- | ----- | -------: | ---: |
> | Dutch	| BERT  	| large |	    4.3 |  9.6 |
> | Dutch	| RoBERTa   	| large |	  13.4 |  7.8 |
> | Dutch	| DeBERTaV3 	| base |  24.1 |  8.1 |
> | Dutch	| DeBERTaV3 	| large |  38.0 | 10.8 |
> | Mutli	| BERT  	| large |    2.8 |  8.1 |
> | Multi	| DeBERTaV3 	| large |  36.4 |  8.6 |
>
> Finally, we will implement the presentation improvement suggestions or add clarification where needed, thanks for your suggestions. For (1), we report previous results if previous results exist on the exact datasets that we use. These do not always exist since we introduce new tasks and new cross-validation splits. We will standardize the descriptions to similar structures across tasks, and clarify the presence/absence of some details in the introduction of Section 2. We will fix the underlining, add the overall summary table and clarify the categories.

---

### Official Review · Reviewer_2NuP · 2023-08-04

**Soundness:** 4

**Excitement:**

4: Strong: This paper deepens the understanding of some phenomenon or lowers the barriers to an existing research direction.

**Paper Topic And Main Contributions:**

This paper presents a new benchmark for Dutch including a variety of nine tasks and introduces the use of relative error reduction (instead of mean scores) relative to a baseline to make future evaluations on this benchmark (more) comparable.

**Questions For The Authors:**

In section 3.2 you list the contents of the benchmarks, but I miss the QA task you mention in section 2.4. Did you not do any fine-tuning for QA?

**Reasons To Accept:**

The work presented in this paper is relevant to the further development of Dutch and multi-lingual LMs. The benchmark is well constructed, tested quite extensibly with existing LMs and the results reveal directions for further research. The paper is well written and has a logical structure.

**Reasons To Reject:**

I currently do not see any reasons to reject this paper.

**Reproducibility:**

5: Could easily reproduce the results.

**Reviewer Confidence:**

4: Quite sure. I tried to check the important points carefully. It's unlikely, though conceivable, that I missed something that should affect my ratings.

**Typos Grammar Style And Presentation Improvements:**

line 005: include -> includes

line 009: models -> maybe language models?

line 043: course -> coarse (occurs a few times in the article. Please change all.)

line 050/051: reference De Bruyne et al contains first names. Please change.

line 140: having a different order of -> having different orders of

line 148: Simple instructions do download -> to download

line 163: CGN -> acronyms has not yet been introduced and might not be familiar to a reader with a non-Dutch NLP background.

Please check capitalisation in the references, e.g. lines 905ff "bert" or lines 945ff "winograd".

---

> ### Author Rebuttal · Authors · 2023-08-29
>
> We want to thank you for complimenting the quality of our paper.
>
> Section 3.2 is indeed missing a mention of QA (extractive question answering) in the enumeration of fine-tuning methods. An earlier draft of this work did not contain the QA task, and we have forgotten to change this line. The relevant QA hyperparameters are actually already included in the appendix and the fine-tuning implementation is present in our codebase. Thanks for pointing this out, we will change the text to reflect this.
>
> Finally, we would like to thank you for pointing out some typos and capitalisation issues. We will of course correct those, and search for remaining errors.

---

### Official Review · Reviewer_ourb · 2023-08-04

**Soundness:** 4

**Excitement:**

4: Strong: This paper deepens the understanding of some phenomenon or lowers the barriers to an existing research direction.

**Paper Topic And Main Contributions:**

The paper presents a set of datasets for nine different tasks that could work as a benchmark for testing Dutch language models, similar to GLUE or other benchmark datasets existing for other languages, but with different tasks. The authors describe the included tasks and their corresponding datasets (some already existed, some were created by the authors), and perform extensive experiments with many available Dutch, and some English and multilingual models. They present their results as RER (Relative Error Reduction) instead of giving the plain metrics for each task, defined as the relative difference in metric compared to a strong baseline, in this case using BERTje model as the strong baseline. It would have been nice, however, to also see the actual metrics of the models, or perhaps present the metrics for this BERTje baseline so we can know what the others are comparing to. There is an interesting analysis of the tasks as the correlation of the metrics for the different systems, the authors find high correlations in similar tasks (e.g. word level tasks), but also discover an interesting correlation between POS and Abusive Language Detection, which might be attributed to the systems deriving their responses only from word level information.


**Reasons To Accept:**

The benchmark datasets are an interesting contribution to the analysis of the Dutch language.

The paper includes extensive experimentation and meaningful analysis of existing models according to the tasks.

The inclusion of the Abusive Language Detection is of particular importance, as it is not included in GLUE or similar benchmarks.


**Reasons To Reject:**

The work might be significant only to researchers working on Dutch, but even then other researchers might find interesting ideas in this work to apply to other languages.


**Reproducibility:**

4: Could mostly reproduce the results, but there may be some variation because of sample variance or minor variations in their interpretation of the protocol or method.

**Reviewer Confidence:**

3: Pretty sure, but there's a chance I missed something. Although I have a good feel for this area in general, I did not carefully check the paper's details, e.g., the math, experimental design, or novelty.

---

> ### Author Rebuttal · Authors · 2023-08-29
>
> We want to thank you for recognizing the value of our work and for pointing out that our work would be valuable for similar work in other languages. We have indeed attempted to motivate and explain our aggregation and data creation methods in a way that is reproducible for other languages.
>
> As you point out, we focus on Relative Error Reduction (RER) scores for our model comparisons, but you indicate that you would have preferred to also see the standard metrics for each task. However, in Table 2 we do already report all standard metric scores next to the RER scores for each model. Moreover, for each task we have described the used metrics and achieved performances of previous works in Section 2. We will clarify in the results section (3.4) that we include these metrics.

---

### Meta-Review · Area_Chair_sBHm · 2023-09-17

**Recommendation:** 4

**Metareview:**

This paper presents a new benchmark for Dutch including a variety of nine tasks and introduces the use of relative error reduction (instead of mean scores) relative to a baseline to make future evaluations on this benchmark (more) comparable.

Reasons To Accept:
- The work presented in this paper is relevant to the further development of Dutch and multi-lingual LMs. The benchmark is well constructed, tested quite extensibly with existing LMs and the results reveal directions for further research. The paper is well written and has a logical structure.
- The paper includes extensive experimentation and meaningful analysis of existing models according to the tasks.
- The inclusion of the Abusive Language Detection is of particular importance, as it is not included in GLUE or similar benchmarks.

Reasons To Reject:
- The work might be significant only to researchers working on Dutch, but even then other researchers might find interesting ideas in this work to apply to other languages.

---

### Decision · Program_Chairs · 2023-10-07

**Decision:**

Accept-Main

**Comment:**

This paper presents a new benchmark for Dutch including a variety of nine tasks and introduces the use of relative error reduction (instead of mean scores) relative to a baseline to make future evaluations on this benchmark (more) comparable.

Reasons To Accept:
- The work presented in this paper is relevant to the further development of Dutch and multi-lingual LMs. The benchmark is well constructed, tested quite extensibly with existing LMs and the results reveal directions for further research. The paper is well written and has a logical structure.
- The paper includes extensive experimentation and meaningful analysis of existing models according to the tasks.
- The inclusion of the Abusive Language Detection is of particular importance, as it is not included in GLUE or similar benchmarks.

Reasons To Reject:
- The work might be significant only to researchers working on Dutch, but even then other researchers might find interesting ideas in this work to apply to other languages.